# Liposomes as a Nanoplatform to Improve the Delivery of Antibiotics into *Staphylococcus aureus* Biofilms

**DOI:** 10.3390/pharmaceutics13030321

**Published:** 2021-03-02

**Authors:** Magda Ferreira, Sandra N. Pinto, Frederico Aires-da-Silva, Ana Bettencourt, Sandra I. Aguiar, Maria Manuela Gaspar

**Affiliations:** 1Centre for Interdisciplinary Research in Animal Health (CIISA), Faculty of Veterinary Medicine, Universidade de Lisboa, 1300-477 Lisboa, Portugal; mscferreira@ff.ulisboa.pt (M.F.); fasilva@fmv.ulisboa.pt (F.A.-d.-S.); 2Faculty of Pharmacy, Research Institute for Medicines (iMed.ULisboa), Universidade de Lisboa, 1649-003 Lisboa, Portugal; asimao@ff.ulisboa.pt; 3Department of Bioengineering, iBB-Institute for Bioengineering and Biosciences, Instituto Superior Técnico, Universidade de Lisboa, 1049-001 Lisboa, Portugal; sandrapinto@ist.utl.pt

**Keywords:** *Staphylococcus aureus*, biofilms, liposomes, infection, transwell model

## Abstract

*Staphylococcus aureus* biofilm-associated infections are a major public health concern. Current therapies are hampered by reduced penetration of antibiotics through biofilm and low accumulation levels at infected sites, requiring prolonged usage. To overcome these, repurposing antibiotics in combination with nanotechnological platforms is one of the most appealing fast-track and cost-effective approaches. In the present work, we assessed the potential therapeutic benefit of three antibiotics, vancomycin, levofloxacin and rifabutin (RFB), through their incorporation in liposomes. Free RFB displayed the utmost antibacterial effect with MIC and MBIC_50_ below 0.006 µg/mL towards a methicillin susceptible *S. aureus* (MSSA). RFB was selected for further in vitro studies and the influence of different lipid compositions on bacterial biofilm interactions was evaluated. Although positively charged RFB liposomes displayed the highest interaction with MSSA biofilms, RFB incorporated in negatively charged liposomes displayed lower MBIC_50_ values in comparison to the antibiotic in the free form. Preliminary safety assessment on all RFB formulations towards osteoblast and fibroblast cell lines demonstrated that a reduction on cell viability was only observed for the positively charged liposomes. Overall, negatively charged RFB liposomes are a promising approach against biofilm *S. aureus* infections and further in vivo studies should be performed.

## 1. Introduction

*Staphylococcus aureus* infections are a major concern in medical care, being a significant cause of morbidity and mortality worldwide. This pathogen is a gram-positive commensal, opportunistic and life-threating bacterium, renowned for its ability to evade the immune system causing systemic infections [1,2]. *S. aureus* is responsible for most cases of hospital-acquired infections in surgical sites, skin and soft tissues, leading to localized diseases such as osteomyelitis and endocarditis, or even bacteremia [1,2].

*S. aureus* strains have the ability to form biofilm organized bacteria, making the eradication of this type of infection extremely difficult [3,4,5,6,7,8]. Bacterial biofilms are composed of bacteria aggregates, involved in an extracellular matrix, containing proteins, polysaccharides and DNA [9]. This structure provides an excellent mechanism of defense, against immune response and blocks the antibiotic penetration, leading to extended hospitalization periods and high social costs [8,10,11,12].

Current therapies involve prolonged administration of high doses of antibiotics. Yet, the sub-therapeutic levels of the antibiotics at infection sites, due to unfavorable pharmacokinetics and reduced penetration through biofilm, are the origin of drug resistance emergence and consequently to antibacterial failure in the clinic, thus representing a major reason of concern [1,11]. The inefficiency of existing therapies in clinical use has led to an intensive research for developing novel therapeutic strategies, aiming to enhance the interaction with bacterial biofilms thus improving the efficacy and safety of antimicrobial agents. Although the discovery of new antibiotics is an obvious approach, it is a time consuming and expensive process, with low levels of success [13,14]. Taking this into account, one of the most explored strategies is repurposing existing antimicrobial compounds and associating them to nanotechnological platforms to get fast-track, approved and more effective therapies.

Drug delivery systems play an important role to improve the safety and efficacy of loaded antibiotics, by allowing more appropriated pharmacokinetics and biodistribution profiles, thus improving the success of antimicrobial compounds already in clinical use [14]. Indeed, several nanotechnological platforms have been developed over the past decades, aiming to deliver the antibiotics at the infection sites and increase their penetration into biofilm structures [10,11].

Among the available nanotechnological platforms, lipid-based nanosystems, particularly liposomes, represent an attractive nanocarrier for the delivery of antibiotics as demonstrated in vitro and in vivo for a wide range of pathogenic organisms, including biofilm associated bacterial pathogens [11,15].

Liposomes are lipid vesicles composed of one or more bilayers enclosing one or various inner aqueous compartments, being able to incorporate both hydrophilic and hydrophobic compounds [11,16,17,18]. This system presents several advantages, such as biocompatibility, easy preparation of sterile formulations and administration through different routes including parenteral and local injection at infected sites [16,17,18,19,20,21]. Moreover, this type of carrier prevents the antibiotic degradation following in vivo administration. The most appealing feature of liposomes is their ability to mimic biological membranes and, by controlling their lipid composition, enabling their fusion within bacterial cell wall, promoting the release of loaded antibiotic that otherwise would not be possible for the compound in free form [11,22,23]. Ultimately, this approach can reduce the number of administrations to exert antibacterial effect, thus enhancing the patient’s quality of life [11].

In this context, the current study aimed to go a step further in the development of antibiotic-loaded liposomes effective against *S. aureus* biofilms. Three antibiotics were chosen for an initial screening, regarding susceptibility tests and incorporation in liposomes. The selected antibiotics were the fluoroquinolone, levofloxacin (LEV), usually recommended for the treatment of *S. aureus* bone and joint infections [24], the glycopeptide vancomycin (VCM), the gold standard antibiotic in clinical practice against *S. aureus* infections [12,25], and the rifamycin, rifabutin (RFB), a broad-spectrum antibiotic covering *S. aureus* strains [26]. To achieve our goal, the design and characterization of liposomes with different lipid compositions for each antibiotic were accomplished and the susceptibility of a *S. aureus* methicillin-susceptible strain (MSSA), both in planktonic and biofilm forms, was assessed. Furthermore, the influence of lipid composition on biofilm interaction, as well as its important role on delivering loaded antibiotic has also been evaluated.

## 2. Materials and Methods

### 2.1. Reagents

Levofloxacin (LEV) and vancomycin (VCM) were obtained from Sigma–Aldrich (St. Louis, MO, USA), and Rifabutin (RFB) from Pharmacy Biotech AB (Uppsala, Sweden). The pure phospholipids, dimyristoyl phosphatidyl choline (DMPC), dimyristoyl phosphatidyl glycerol (DMPG), dipalmitoyl phosphatidyl choline (DPPC), dipalmitoyl phosphatidyl glycerol (DPPG) and dioleoyl phosphatidyl ethanolamine (DOPE) were purchased from Lipoid (Ludwigshafen, Germany). Stearylamine (SA) was obtained from Sigma–Aldrich (St. Louis, MO, USA). Cholesteryl hemisuccinate (CHEMS) and rhodamine covalently linked to phosphatidylethanolamine (Rho-PE) were purchased from Avanti Polar Lipids (Alabaster, AL, USA). Nuclepore Track-Etch Membranes were purchased from Whatman Ltd. (Maidstone, UK). Thiazolyl Blue Tetrazolium Bromide (MTT) and Crystal Violet (CV) were obtained from Panreac Applichem, ITW Reagents (Darmstadt, Germany). Culture media, Mueller-Hinton Agar (MHA) and Mueller-Hinton Broth (MHB) were obtained from Oxoid, Ltd. (Basingstoke, UK), Tryptic Soy Broth (TSB) from Biokar (Pantin, France) and Columbia Agar + 5% sheep blood (CA) from bioMérieux (Marcy l’Étoile, France). The SYTO 9 was obtained from Molecular Probes (Eugene, OR, USA). All the remaining chemicals used were of analytical grade.

### 2.2. Preparation of VCM- and LEV-Loaded Liposomes

VCM and LEV liposomes composed of the selected phospholipids were prepared by dehydration-rehydration technique, using a passive method [16,17,27]. Briefly, the phospholipids at an initial lipid concentration of 30 µmol/mL were dissolved in chloroform and the solvent was evaporated in a rotary evaporator (Buchi R-200, Flawil, Switzerland) to obtain a thin lipid film in a round-bottom flask. The obtained lipid film was dispersed with the respective antibiotic solution (LEV—1 mg/mL or VCM—2 mg/mL) and the so-formed suspensions were frozen (−70 °C) and lyophilized overnight (freeze-dryer, CO, USA). The rehydration of the lyophilized powder was performed with HEPES buffer, pH 7.4 (10 mM HEPES, 140 mM NaCl), in two steps, to enhance the antibiotics incorporation: first, rehydration was done up to two-tenth of the volume of the original dispersion, and 30 min after that, rehydration was completed up to the starting volume. The rehydration steps were always performed at a temperature above the phase transition temperature (Tc) of the phospholipids used. To homogenize and reduce the mean size of liposomes, the so-formed suspensions were submitted to an extrusion step, through polycarbonate membranes with defined pore size until reaching lipid vesicles with approximately 100 nm of diameter, under a nitrogen pressure of 100–500 IB/in^2^, using an extruder device (Lipex: Biomembranes Inc., Vancouver, BC, Canada). The non-incorporated antibiotics were separated by ultracentrifugation at 250,000× *g*, for 2 h at 15 °C in a Beckman LM-80 ultracentrifuge (Beckman Instruments, Inc., Fullerton, CA, USA). The pellet was finally suspended in HEPES buffer, pH 7.4.

### 2.3. Preparation of RFB-Loaded Liposomes

RFB was loaded in pre-formed empty liposomes following the establishment of an ammonium sulphate gradient between the intraliposomal and extraliposomal media according to Gaspar et al. [18]. Briefly, the phospholipids were dissolved in chloroform and the solvent was evaporated in a rotary evaporator (Buchi R-200, Flawil, Switzerland) to obtain a thin lipid film in a round-bottom flask. The obtained lipid film was dispersed in deionized water, at a phospholipid concentration of 30 µmol/mL. The so-formed lipid suspensions were frozen (−70 °C) and lyophilized overnight (freeze-dryer, Edwards, CO, USA). The rehydration of the lyophilized powder was performed in a buffer solution containing 125 mM ammonium sulphate, pH 5.0. The temperature of the hydrating medium was above Tc of the phospholipids used. To homogenize and reduce the mean size of liposomes, the dispersions were sequentially filtered, through polycarbonate membranes with defined pore size, until reaching a diameter of approximately 100 nm, under a nitrogen pressure of 100–500 IB in^−2^, using an extruder device (Lipex: Biomembranes Inc., Vancouver, BC, Canada). An ammonium sulphate gradient was created by replacing the extraliposomal medium with HEPES buffer, pH 6.9, using a desalting column (Econo-Pac^®^ 10 DG; Bio-Rad Laboratories, Hercules, CA, USA). A RFB solution (at a concentration of 0.5 mg/mL) prepared in the HEPES buffer, pH 6.9, was then incubated with unloaded liposomes, for 1 h under stirring and at a temperature higher than the Tc of the respective lipid mixture. The non-incorporated RFB was separated by ultracentrifugation at 250,000× *g*, for 2 h at 15 °C in a Beckman LM-80 ultracentrifuge (Beckman Instruments, Inc., Fullerton, CA, USA). The pellet was finally suspended in the previous mentioned buffer [18]. Unloaded liposomes were also prepared by the same method. Fluorescent liposomes were prepared by adding rhodamine covalently linked to phosphatidylethanolamine (Rho-PE) to the lipid mixture at 0.2 mol% [28].

### 2.4. Liposomes Characterization

Liposomes were characterized in terms of incorporation parameters, namely antibiotic and lipid contents. LEV was quantified spectrophotometrically (Shimadzu 160-A, Shimadzu Corporation, Kyoto, Japan), at 300 nm, after disruption of the liposomes with ethanol [18]. VCM quantification was performed by Lowry and Folin technique [29,30,31]. The phosphate content of all liposomal formulations was determined using a colorimetric method as described by Rouser et al. [32].

The incorporated RFB in liposomes was determined by high-performance liquid chromatography (HPLC) according to Gaspar et al. [18]. The HPLC system is based on a System Gold (Beckman Instruments, Inc., Brea, CA, USA), a Midas Spark 1.1 autoinjector and a Diode-Array 168 detector (Beckman Instruments, Inc., Brea, CA, USA). The detector wavelength was set to 275 nm. The analytical column was a LiChroCART^®^ (125-4), Purospher^®^ Star RP-C8 (5 µm) (Merck, Darmstadt, Germany). The mobile phase consisted of 0.05 M potassium hydrogen phosphate and 0.05 M sodium acetate (pH 4.0, adjusted with acetic acid):acetonitrile (53:47, *v*/*v*) with a flow rate of 1 mL/min at 25 °C. A stock solution of RFB (125 µg/mL) was prepared in the mobile phase. Further standard solutions were made by diluting the initial stock RFB solution in the mobile phase. Three eight-point calibration curves ranging from 1 to 25 µg/mL, with a loop of 40 µL, were used for the quantification of RFB samples. A standard solution of 5.0 µg/mL was stored at −30 °C and a sample of this standard solution was always injected together with the analyzed samples to verify the precision of the obtained RFB concentration in samples and controls from their peak area concentration response.

Liposomes were characterized in terms of lipid composition and by the following incorporation parameters: initial and final antibiotic (AB) to lipid ratios ((AB/Lip)i and (AB/Lip)f, respectively) and incorporation efficiency (I.E.) in %, defined as Equation (1):(1)$$I.E.\left(\%\right)=\frac{\left(\frac{\mathrm{AB}}{\mathrm{Lip}}\right)f}{\left(\frac{\mathrm{AB}}{\mathrm{Lip}}\right)i}\times 100$$

Liposomes mean size was determined by dynamic light scattering in a Malvern Zetasizer Nano-S (Malvern Instruments, Malvern, UK) at a standard laser wavelength of 663 nm. The system also reports polydispersity index (P.I.) as a measure of particle size distribution, ranging from 0.0 for an entirely monodisperse sample up to 1.0 for a polydisperse suspension. The zeta potential was calculated by laser doppler spectroscopy in a Malvern Zetasizer Nano-Z (Malvern Instruments, Malvern, UK).

### 2.5. Bacterial Strain and Culture Conditions

A methicillin susceptible *S. aureus* ATCC^®^25923™ (MSSA) was obtained from the American Type Culture Collection (ATCC; Manassas, VA, USA). Bacterial stocks were prepared from overnight cultures on MHA, in MHB with 20% of glycerol at 37 °C and stored at −80 °C until further use. For each assay fresh cultures were prepared from frozen stocks on MHA or CA incubated overnight at 37 °C.

### 2.6. Planktonic S. aureus Susceptibility to Antibiotics

The antibacterial activity of free antibiotics and RFB-loaded liposomes was evaluated by the broth microdilution method, according to guidelines of the Clinical and Laboratory Standards Institute [33], followed by turbidity evaluation. Briefly, the formulations were diluted in MHB to produce a 2- or 4-fold dilution with concentrations ranging from: 0.004 to 1.000 µg/mL in case of LEV; 0.195 to 6.250 µg/mL for VCM and 0.002 to 0.200 µg/mL for RFB. Bacterial suspensions were performed from a MSSA overnight culture diluted in MHB until reaching a value of 0.5 in a McFarland scale equivalent to 10^8^ colony forming unit per mL (CFU/mL), by measuring the optical density (OD) at 600 nm. Bacterial suspension was placed in 96-well cell culture plates at 5 × 10^5^ CFU/mL and incubated with the antibiotics or the formulations, at 37 °C during 24 h, under static conditions. A negative control containing a suspension of bacteria in MHB, without antibiotic, and a sterile control containing only MHB, were performed in parallel. Minimum inhibitory concentration (MIC) was determined spectrophotometrically, at 570 nm in an iMark™ microplate absorbance reader (Bio-Rad laboratories, Inc., Hercules, CA, USA) and it was defined as the lowest antibiotic concentration able to prevent visible bacterial growth, resulting in the absence of turbidity.

### 2.7. S. aureus Biofilm Susceptibility to Antibiotics

Overnight cultures were incubated at 10^6^ CFU/mL in 96-well cell culture plates with TSB supplemented with 0.25% of glucose (TSB 0.25%). Biofilms were formed under static conditions for 24 h at 37 °C. After biofilm growth, the content of each well was washed twice with a sterile solution of phosphate buffered saline (PBS) to remove non-adherent bacteria. Serial dilutions of free antibiotics and RFB-loaded liposomes performed in TSB 0.25%, were added to each well. Studied concentrations ranged from: 0.098 to 100.000 µg/mL for LEV; 6.250 to 200.000 µg/mL for VCM and 0.001 to 10.000 µg/mL for RFB in free or liposomal forms. Plates were incubated for 24 h at 37 °C. The bacterial cell viability and the biofilm biomass were measured by the MTT reduction assay and CV staining, respectively, after carefully rinsing twice the attached bacterial cells with sterile PBS. Results were confirmed by CFU quantification.

MTT reduction assay was performed as previously described by Brambilla et al. [34] with some exceptions. After biofilm rinsing, 200 µL of a 125 µg/mL MTT solution in PBS was added and incubated at 37 °C during 2 h. Then MTT solution was removed and replaced by dimethyl sulfoxide (DMSO) to dissolve the MTT formazan product. The OD of samples was measured at 570 nm using an iMark^TM^ microplate absorbance reader (Bio-Rad laboratories, Inc., Hercules, CA, USA). Bacterial cells recovered of biofilm structure were evaluated by determining the percentage of viable bacterial cells (tested samples) related to the respective control (CTR, biofilm grown in the presence of TSB 0.25%), which represented 100% of viability. Bacterial cell viability (%) was defined by Equation (2):(2)$$\mathrm{Bacterial}\;\mathrm{cell}\;\mathrm{viability}\;(\%)\;=\;\frac{{\mathrm{OD}}_{t}}{{\mathrm{OD}}_{\mathrm{CTR}}}\times 100$$
where OD_t_ is the optical density of the bacterial cells incubated with the tested formulations and OD_CTR_ is the optical density of the control cells, corresponding to 100% cell viability.

Minimum biofilm inhibitory concentration (MBIC) was determined through the results of the MTT method and was defined as the lowest antibiotic concentration able to inhibit more than 50% of bacterial growth related to untreated controls (MBIC_50_). The determination of MBIC_50_ was performed by sigmoidal fitting analysis considering a confidence level of 95%.

For CV staining method [7,35] the attached bacterial cells were air-dried at room temperature (RT) for 15 min and subsequently stained with 200 µL of a CV solution 0.125% (*w*/*v* in water). After incubation at RT during 15 min, biofilms were washed twice with sterile PBS to remove excess dye, followed by a drying step at RT. Stained biofilms were then dissolved in 200 µL of ethanol and diluted at a ratio of 1:10. The OD was measured at 570 nm using an iMark™ microplate absorbance reader (Bio-Rad laboratories, Inc., Hercules, CA, USA). Mature biofilm grown in the presence of TSB 0.25% was used as control group (CTR) and represented 100%. Biofilm biomass (%) was defined by Equation (3):(3)$$\mathrm{Biofilm}\;\mathrm{Biomass}\;(\%)\;=\;\frac{{\mathrm{OD}}_{t}}{{\mathrm{OD}}_{\mathrm{CTR}}}\times 100$$
where OD_t_ is the biofilm biomass of the bacterial cells incubated with the tested formulations and OD_CTR_ is the optical density of the control biofilm biomass, corresponding to 100% biofilm biomass.

CFU counts were performed as previously described, with some modifications [36]. Biofilm bacterial cells were suspended in 200 µL of PBS by vigorous pipetting, followed by 10-fold dilutions. Ten µL of selected dilutions were drop-plated on the MHA. CFU were counted after 24 h of incubation at 37 °C. In all methods, negative control of biofilm (without antibiotic) and sterile control (only medium) were included.

### 2.8. Influence of Lipid Composition on S. aureus Biofilm Interaction

Unloaded liposomes with selected lipid compositions were prepared by including Rho-PE at 0.2 mol% related to total lipid. *S. aureus* biofilm was cultured as described above (Section 2.7). After biofilm growth, the medium was removed, and each well was carefully washed twice with sterile PBS. Next, 200 µL of rhodamine-labeled liposomes at different lipid concentrations (0.5, 1.0, 2.0, 2.5, 5.0 µmol/mL) prepared in TSB 0.25%, were added to the biofilm. After 2, 4 or 24 h of incubation at 37 °C, the medium was removed, and the biofilm was washed twice with sterile PBS. The biofilm was suspended in 400 µL of ethanol. Liposomes associated to biofilm structure were measured by spectrofluorimetry (FLUOstar OPIMA BGM Labtech) (λ excitation = 540 nm; λ emission = 620 nm). Calibration curves were performed for each formulation tested at a lipid concentration ranging from 0.02 to 0.25 µmol/mL in ethanol.

### 2.9. In Vitro Evaluation of RFB Formulations in a Biofilm Transwell Model

The experiment was performed as previously described by Harriott and colleagues [37], with some modifications. Biofilm was grown in a porous membrane cell culture insert (designed by apex) (porous diameter of 0.4 µm; Greiner Bio-One North America, Inc., Monroe, USA) in TSB 0.25%, placed on 24-well cell culture plate with 500 µL of the same culture medium per well (designed by base), during 24 h at 37 °C. After biofilm formation, the medium was removed, the biofilm was carefully washed one time with sterile PBS and fresh medium was replaced in the base. RFB-loaded and unloaded in rhodamine-labeled liposomes at a lipid concentration of 0.5 µmol/mL were added to the mature biofilm formed in the apex and incubated for 24 h at 37 °C. In addition, free RFB at 1 µg/mL was also tested for comparison with rhodamine-labeled liposomes. Fluorescence intensity of the samples recovered from the apex diluted in ethanol were determined by spectrofluorimetry, as described in Section 2.8. The biofilm integrity was evaluated by the CV method described in Section 2.7.

### 2.10. In Vitro Evaluation of RFB Formulations by Confocal Scanning Laser Microscopy

The influence of lipid composition in *S. aureus* biofilm interaction was analyzed by confocal scanning laser microscopy (CSLM). For this, unloaded and RFB-loaded rhodamine-labeled liposomes, prepared with different lipid compositions as described in Section 2.9, were used. Biofilms were grown overnight in TSB 0.25% in 8-well chambered coverslips (Ibidi GmbH, Munich, Germany) according with Pinto et al. [38]. The next day culture medium was removed, and the biofilms were incubated with liposomes in TSB 0.25% at a lipid concentration of 1.5 µmol/mL, for 4 h at 37 °C. After that, biofilms were washed with PBS and then stained in the dark at RT with 3 µM SYTO 9 (30 min). Untreated biofilms were used as negative control. The biofilms were visualized using a Leica TCS SP5 inverted microscope (Leica Mycrosystems CMS GmbH, Mannheim, Germany) equipped with a continuous Ar ion laser (Multi-line LASOS^®^ LGK 7872 ML05). Images were collected at 512 by 512 pixels and using a scan rate of 100 Hz per frame. A 63x1.2 N.A. water immersion objective was used (HCX PL APO CS 63.0 × 1.20 WATER UV). SYTO-9 images were recorded using a 488 nm excitation line and emission was collected at 501–570 nm. Rho-PE images were recorded using 514 nm excitation line and emission was collected at 610–760 nm (to minimize a possible crosstalk between SYTO-9 and Rho). For each condition different areas in biofilm and different plans (xzy, xyz) were measured. Data analysis was carried on using ImageJ software.

### 2.11. Cell Lines Culture Conditions

Human MG-63 osteoblast (ATCC^®^CRL-1427™) and mouse L929 fibroblast (ATCC^®^CCL-1™) cell lines were obtained from ATCC (Manassas, VA, USA) and cultured in Dulbecco’s Modified Eagle’s medium (DMEM) with high glucose (4500 mg/L), supplemented with 10% fetal bovine serum, 100 IU/mL of penicillin and 100 µg/mL streptomycin (hereafter designated as complete medium), at 37 °C under 5% CO_2_ atmosphere. Maintenance of cultures was accomplished every 2–3 days, until cells reached a confluence of about 80%.

### 2.12. Preliminary In Vitro Safety Assessment of Liposomes

Cell viability of human MG-63 osteoblast and mouse L929 fibroblast were evaluated in the presence of unloaded or RFB-loaded liposomes by MTT reduction assay [39]. Cells were seeded, at a concentration of 5 × 10^4^ cells/mL, in 96-well plates (200 µL) and were incubated for 24 h, in culture conditions described in Section 2.11. After this, medium was removed and cells were incubated with unloaded or RFB-loaded liposomes, prepared with different lipid compositions, at a lipid concentration of 0.5, 1.0, 2.0 and 5.0 µmol/mL. At 24 h after incubation, medium was discarded, cells were washed with PBS and 50 µL of MTT solution at 0.5 mg/mL in incomplete medium was added. Following an incubation period of 3 h at 37 °C, 100 µL of DMSO was added to each well, to solubilize the formazan crystals and the OD was measured at 570 nm in a microplate reader Model 680 (Bio-Rad, CA, USA). Cell viability was evaluated by determining the percentage of viable cells (tested formulations) related to the respective control (CTR, cells incubated with complete medium), as defined by Equation (4).
(4)$$\mathrm{Cell}\;\mathrm{viability}\;(\%)\;=\;\frac{{\mathrm{OD}}_{t}}{{\mathrm{OD}}_{\mathrm{CTR}}}\times 100$$
where OD_t_ is the optical density of the cells incubated with the tested formulations and OD_CTR_ is the optical density of control cells, corresponding to 100% cell viability.

### 2.13. Statistical Analysis

All results are expressed as mean ± standard deviation (SD). Significant differences between experimental groups were assessed by a one-way or two-way analysis of variance (ANOVA), with Tukey’s or Bonferroni’s multiple comparisons post-hoc test, using GraphPad Prisma^®^8 (GraphPad Software, San Diego, CA, USA). For all analysis, differences between groups were considered statistically significant when *p* < 0.05.

## 3. Results and Discussion

*S. aureus* biofilm infections represent a huge therapeutic challenge due to the poor antibiotic penetration into the complex biofilm microenvironment [11,40]. Therefore, new drug delivery strategies that enable high biofilm penetration and effective bacterial clearance are urgently needed. A promising approach is the use of nanoparticle delivery platforms for antibiotic delivery. Their flexibility in terms of composition, compound incorporation and functionalization chemistry may allow the development of potent nanotherapies by promoting a higher biofilm interaction and in situ release of loaded antibiotics [41]. Among the wide panel of nanoplataforms, liposomes constitute one of the most promising nanoparticle-based technologies for biofilm infections [42,43,44]. Indeed, their excellent biocompatibility, controlled release of loaded antibiotics and biofilm penetrating features, have been demonstrated to improve biofilm clearance for a variety of bacterial pathogens as reviewed by Rukavina et al. [23]. In the present study we aimed to develop liposomes incorporated with clinically used antibiotics and investigate the influence of the lipid compositions in the penetrability, antibiotic release and in vitro clearance of *S. aureus* biofilms.

### 3.1. Planktonic and Biofilm S. aureus Susceptibility to Antibiotics

VCM and fluoroquinolones, such as LEV, alone or in combination with rifamycin class antibiotic are the most common antimicrobial therapies for biofilm-associated *S. aureus* infections [45]. Among the rifamycin class of antibiotics, rifampicin is the most commonly used, although with severe adverse side effects due to gastrointestinal intolerance or drug-drug interactions. As such, other rifamycins, such as RFB are currently being evaluated as a possible anti-biofilm therapy due to their favorable toxicity and drug interaction behavior [26,46]. Aiming to choose the antibiotic with higher antimicrobial activity against planktonic and biofilm MSSA strain, susceptibility tests were performed for the three selected antibiotics RFB, LEV and VCM (Figure 1 and Appendix A [47,48,49,50,51,52]).

The susceptibility of *S. aureus* ATCC strain to the three antibiotics was assessed in planktonic cells by the broth microdilution method as depicted in Figure 2a–c. Following previous selection of the most appropriated range of concentrations for the three antibiotics, the MIC determination for RFB, VCM and LEV (Table 1) revealed that bacterial growth inhibition was observed at concentrations above 0.006, 1.562 and 0.125 µg/mL, respectively, with RFB presenting an in vitro antibacterial activity 260-fold higher than the gold standard VCM. Rifampicin was evaluated in a previous study, conducted by Grohs et al., against the same strain, in which a MIC value 2.5-fold higher (0.015 µg/mL) than the one obtained for RFB was observed [53], further supporting the superior efficacy of RFB. The results obtained for VCM and LEV were in agreement with what is described in the literature for MSSA strains, around 1 µg/mL [54,55] and ranging from 0.10 to 0.78 µg/mL [56,57], respectively.

Antibiotic activity is normally impaired by biofilm structured bacteria with very few antibiotics capable of maintaining similar antibacterial potential against planktonic and biofilm forms. As observed by Mandell J. et al. [58], who compared the activity of 10 antibiotics in mature and planktonic biofilms, most of them presented a 2–3 log increase of the MBIC, with the exception for daptomycin that displayed only an augment of 1-log [58]. For this reason, it is important to evaluate the antibiotics activity in both planktonic and biofilm forms.

As such, the mature MSSA biofilm strain susceptibility to the three antibiotics was evaluated by bacterial viability and biofilm biomass quantifications, as presented in Figure 2d–f and Figure 2g–i, respectively. For RFB, cell viability below 50% was observed for concentrations above 0.005 µg/mL (Figure 2d) that were also corroborated by biofilm biomass results (Figure 2g). In the case of LEV, a 4-fold higher concentration (6.250 µg/mL) was needed to attain a similar effect (Figure 2f,i), while VCM, at a 20-fold higher concentration (200 µg/mL), did not exhibit the antibacterial effect observed for RFB (Figure 2e,h). A comparison of the MIC and MBIC_50_ for each antibiotic is presented in Table 1. As it can be observed, MBIC_50_ for VCM was significantly higher than the respective MIC (MBIC_50_ > 200.000 µg/mL), while for LEV it was 76-fold higher (MBIC_50_ of 9.468 µg/mL), clearly demonstrating the well-known biofilm induced resistance effect comparatively to the respective planktonic counterpart [59]. Surprisingly, RFB presented a minimum inhibitory concentration for planktonic bacterium similar to the value obtained to inhibit 50% of biofilm growth (MIC = 0.006 µg/mL vs MBIC_50_ = 0.005 µg/mL). To confirm the higher in vitro activity of RFB in biofilm structured *S. aureus*, the analysis of viable bacteria reduction by CFU counts for two antibiotic concentrations, 0.2 and 0.8 µg/mL, was performed (Figure 2j,k). A 3-log reduction of viable bacteria in the presence of RFB was observed for the lowest concentration, in comparison to the control (CTR, biofilm without antibiotic) with this effect being potentiated to a 4-log reduction at the highest concentration tested. For VCM and LEV no differences were registered for the two concentrations tested that displayed less than a 1-log reduction when compared to CTR. This data attests the potent antimicrobial activity of RFB, in the free form, against *S. aureus* biofilms, in comparison with the antibiotics clinically used. Indeed, the comparable activity of RFB against planktonic and biofilm of *Staphylococcus* species was recently described in a case study by Doub J. et al. [46], in which RFB was used instead of rifampin for the treatment of 10 patients with confirmed staphylococcal biofilm infections, further supporting the potential of this antibiotic against *S. aureus* biofilms associated infections [46]. Despite the promising therapeutic efficacy of RFB in vitro, high antibiotic doses are necessary to be administered in vivo to achieve inhibitory concentrations at the infected sites, which could lead to systemic toxicity and development of resistant strains. To overcome systemic toxicity and biofilm penetration issues, we have incorporated the three antibiotics in liposomes and evaluated their antimicrobial and biofilm penetration properties.

### 3.2. Physicochemical Characterization of Antibiotics-Loaded Liposomes

An unprecedent investment in research focusing on engineered nanoparticles has been observed for the last decade with the promise of significant advances in terms of drug targeting, controlled release, improved bioavailability and biocompatibility and ultimately, reduction of toxicity and side effects [60]. Among the extensive panel of nanoparticles available, liposomes distinguish themselves in terms of physicochemical properties and incorporation features that make them efficient antibiotic delivery systems. A major advantage of liposome use for bacterial infections is its intrinsic ability to fuse with phospholipid membranes enabling the release of high concentrations and greatly improving antibiotic drug delivery at infection sites. This effect would not be possible when the antibiotic is tested in the free form [22,23].

Taking this into account, the selection of lipid compositions able to improve the interaction and accumulation within the bacterial biofilm together with an enhanced release of loaded antibiotic in situ must be considered and optimized. Moreover, the particle size, bilayer morphology, surface properties, and encapsulation efficacy are also important characteristics to take into consideration [61].

With this in mind, liposomes incorporating the selected antibiotics were prepared with phospholipids of different Tc, different charges (neutral, negative and positive) or by including lipid constituents with fusogenic properties (DOPE). The main phospholipids used were DMPC (Tc of +23 °C), DMPG (Tc of +23 °C), DOPE (Tc of −16 °C), DPPC (Tc of +41 °C) and DPPG (Tc of +41 °C). Liposome charge is an important characteristic that affects liposome-biofilm interaction and its ability to release the antibiotic. To evaluate the influence of the charge, stearylamine (SA) was selected as a hydrophobic surface modifier to achieve positively charged liposomes, while DPPG and DMPG were included in the lipid composition to obtain negatively charged liposomes. All antibiotics were also incorporated in liposomes with fusogenic properties, since it has been described that this lipid composition increased their capacity to penetrate biofilm structure and fuse with the outer membrane of bacteria, due to the fluidity of the phospholipid bilayer [10,12,62]. In addition, the use of these types of lipids, such as DOPE in combination with CHEMS, promotes the destabilization of the liposome lipid bilayer and stimulates the release of incorporated antibiotics at a lower pH [63]. It is an important feature that could be advantageous, since the biofilm environment is characterized by having a slightly acidic pH [10].

The methodology for the incorporation of the hydrophobic compound RFB in liposomes is well established by our group [18,64]. Therefore, RFB was incorporated by an active method in preformed unloaded liposomes after establishment of an ammonium sulphate gradient [18]. RFB liposomes bilayers were constituted by phospholipids with a moderate fluidity, DMPC or DMPG with a Tc of 23 °C. The use of DMPC as the main phospholipid in RFB-loaded liposomes presents a compromise between drug entrapment and stability as observed in previous studies [10,18]. For the preparation of VCM liposomes, a hydrophilic compound, a passive method using rigid lipids such as DPPC to improve its stability was selected. Indeed, the lipid compositions used for VCM-loaded liposomes have already been used in a previous study by members of the present team [65,66]. Regarding LEV different lipid compositions and preparation methods were tested (Table 2 and Appendix A).

Following liposome development, the influence of the selected lipid compositions on antibiotics incorporation parameters was evaluated. The final antibiotic to lipid ratio (loading capacity) and Incorporation Efficiency (I.E.), as well as mean size and surface charge for all tested lipid compositions and antibiotics under study are depicted in Table 2. RFB-loaded liposomes demonstrated the most suitable results in terms of incorporation parameters particularly for the negatively charged nanoformulations, with loading capacity of 36 and 57 µg per µmol of lipid and I.E. of 51 and 87% for the lipid compositions DMPC:DMPG and DMPC:DOPE:CHEMS, respectively. VCM liposomes presented a loading capacity of 23 and 45 µg per µmol of lipid and an I.E. of 19 and 32%, for the lipid compositions DPPC:DOPE:CHEMS and DPPC:DPPG. For LEV-loaded liposomes no differences in the incorporation parameters were observed for the tested lipid compositions. LEV-loaded liposomes displayed a loading capacity < 2 µg per µmol of lipid and an I.E. below 3%.

The data obtained showed that the fusogenic lipid composition (DMPC:DOPE:CHEMS) presented the highest loading capacity and I.E. for RFB liposomes. The higher fluidity of the lipid composition, as a result of the presence of DOPE, allows a higher penetration through the lipid bilayer of RFB, that is incubated in uncharged state with empty liposomes, being able to cross through the lipid bilayers and once reaching the aqueous compartment gets charged, staying sequestered in the internal aqueous phase. The same methodology has been widely used for efficient and stable active loadings of amphipathic weak bases in liposomes [67]. For the incorporation of the hydrophilic glycopeptide antibiotic, VCM, a passive method was used. Two negatively charged lipid formulations were tested. Again as observed for RFB, the more fluid lipid composition containing DOPE and CHEMS displayed higher loadings in comparison to DPPC:DDPG, phospholipids with a Tc of 41 °C.

Regarding the liposomes mean size, it has been described that smaller vesicles (below 0.2 µm) enhance the therapeutic effect towards biofilm infections [11]. In this study, for the different antibiotic-loaded liposomes the mean size ranged from 0.11 to 0.17 µm with a polydispersity index below 0.15 ensuring a high homogeneity of the liposomes. Formulations including SA showed a positive zeta potential (LEV-DMPC:SA of +18 ± 1 mV and RFB-DMPC:SA of +13 ± 2 mV). A negative zeta potential was observed for the remaining formulations (between −21 to −30 mV) (Table 2). The negatively charged observed for liposomes is in accordance with the lipids used in the lipid composition, namely DMPG, DPPG and CHEMS. From a general point of view, charged liposomes are more stable as the presence of a charge on the surface induces electrostatic repulsion among liposomes allowing also to promote the interaction of liposomes with cells [68].

Several liposomal formulations have been developed to address biofilm associated *S. aureus* infections [11]. In fact, LEV-loaded liposomes was already developed by Zhang X. et al. [69] for the treatment of pulmonary infections. Moreover, Scriboni A. and co-workers [12] also designed VCM-loaded liposomes, using different lipid compositions to achieve an antibacterial activity improvement of VCM against *S. aureus* biofilms. In contrast, to the best of our knowledge, RFB liposomal formulations have not yet been developed and evaluated against *S. aureus* biofilms. However, RFB-liposomes have demonstrated high potential for the treatment of mycobacterial infections [18,64]. Despite these reports, few studies evaluated the influence of lipid composition in liposome-biofilm interaction, particularly in *S. aureus* biofilms, which is the main focus of the present study.

### 3.3. Planktonic and Biofilm S. aureus Susceptibility to RFB-Loaded Liposomes

Based on the obtained results in terms of antibacterial effect and incorporation parameters for the three antibiotics included in the present study, RFB displayed the lowest MIC and MBIC_50_ values in planktonic and biofilm assays and higher loading values in liposomes. Taking these results into account, RFB was the selected antibiotic for further studies, including the assessment of the influence of the lipid composition on the biofilm interaction. To accomplish this, susceptibility assays of RFB liposomes were performed using the following lipid compositions: DMPC:DOPE:CHEMS, DMPC:DMPG and DMPC:SA, hereafter designated by LIP1, LIP2 and LIP3, respectively.

On a first approach, we assessed the influence of RFB-loaded in the three lipid compositions in terms of antibacterial activity in both planktonic and mature *S. aureus* biofilm (Figure 3). For all RFB liposomes, the MIC obtained was 0.006 µg/mL, the same as the one achieved for the antibiotic in the free form as depicted in Figure 3a and Table 3. Statistically significant differences among the tested formulations were observed for the lowest concentration, 0.002 µg/mL (Figure 3a). For this RFB concentration, RFB LIP1 and RFB LIP3 presented statistically significant differences in comparison to free RFB with a *p* value of 0.0026 and 0.0005, respectively. Unloaded liposomes did not influence bacterial growth, indicating that the antibacterial effect was derived from the RFB activity and not from the lipid composition used for liposomes preparation (Appendix A). Data obtained are supported with published studies, which revealed that antibiotics incorporated in liposomes maintained or even improved their antibacterial properties [10,16,18,70].

The influence of the lipid composition used for the preparation of RFB liposomes was also evaluated in mature biofilm assays. Bacterial cell viability and biofilm biomass profile of *S. aureus* biofilm incubated with RFB concentrations ranging from of 0.001 to 10.000 µg/mL were performed (Figure 3b,c). Again, as evidenced for planktonic *S. aureus*, statistically significant differences were observed between RFB formulations, particularly for low concentrations. Bacterial cell viability data revealed a high performance of RFB LIP1 for the lowest concentration tested (0.001 µg/mL), with a statistical difference of *p* = 0.0007 compared to the free RFB and *p* = 0.0294 to RFB LIP3 (Figure 3b). In the mature biofilm, a bacterial cell viability reduction above 50% was achieved for concentrations above 0.002 µg/mL in case of negatively charged liposomes (RFB LIP1 and RFB LIP2) and free RFB. The same reduction behavior was only obtained for positively charged formulation (RFB LIP3) at RFB concentration close to 0.010 µg/mL. The effect of free and liposomal RFB in biofilm biomass is presented in Figure 3c. A biomass reduction above 50% was reached for RFB concentrations higher than 0.002 µg/mL for RFB in free form or loaded in liposomes. In both microbiological assays, biofilm growth was not influenced by the presence of unloaded liposomes (Appendix A).

Bacterial cell viability data provided the determination of MBIC_50_ for each formulation, presented in Table 3. Although, slight differences were observed between RFB in free form or loaded in liposomes no statistically significant differences were found, according to one-way ANOVA (Tukey’s multiple comparisons test) analysis. Free RFB presented a MBIC_50_ of 0.005 µg/mL, while for negatively charged liposomes (RFB LIP1 and RFB LIP2) the values were similar (0.002 µg/mL). LIP3 showed a slightly higher MBIC_50_, 0.006 µg/mL (*p* > 0.05).

### 3.4. Influence of Lipid Composition on S. aureus Biofilm Interaction

RFB incorporated in negatively charged liposomes (RFB LIP1 and RFB LIP2) displayed a slight decrease on the MBIC_50_ value in comparison with RFB in free form or incorporated in positively charged liposomes (RFB LIP3). However, aiming to elucidate the most effective lipid composition in terms of biofilm interaction, and consequent antibiotic delivery, in vitro assays using rhodamine-labelled liposomes were performed. A preliminary assay was carried out to understand the influence of lipid composition and concentration in liposome-biofilm interaction (Appendix A). Here, the three unloaded lipid compositions were incubated at two lipid concentrations (2.5 and 5 µmol/mL) for 2 h with mature MSSA biofilm. The rho-PE incorporation in these formulations did not change liposomes physicochemical properties (Appendix A). In this assay, it was possible to observe a lipid- and concentration-dependent behavior, with LIP3, positively charged, presenting the highest interaction with the *S. aureus* biofilm in both concentrations tested, in comparison to the other two lipid compositions (*p* < 0.001). Subsequently the percentage of lipid associated to biofilm was determined using four different lipid concentrations (0.5, 1.0, 2.0 and 5.0 µmol/mL of lipid) and two time points (4 h (Figure 4b) and 24 h (Figure 4c)). Again, the LIP3 formulation presented higher biofilm-interaction than the LIP1 and LIP2 in all conditions tested. A maximum of 19% and 32% of LIP3 associated to biofilm was achieved following 4 h and 24 h of incubation, respectively, for a lipid concentration of 0.5 µmol/mL. A significant difference with a *p* < 0.0001 for the two negatively charged lipid compositions tested was achieved. For all formulations analyzed, a superior liposome association to biofilm was observed for the lowest lipid concentration studied (0.5 µmol/mL), proving that the lipid concentration used is an important parameter that could also influence the anti-biofilm efficacy. In this way, to maximize the liposome association to biofilm, the lipid concentration of 0.5 µmol/mL and an incubation time of 24 h were the experimental conditions chosen for further assays.

### 3.5. In Vitro Evaluation of RFB Formulations in a Biofilm Transwell Model

One of the key issues regarding new antimicrobial strategies towards biofilm bacteria is the validation of its penetrability or internalization within the whole biofilm structure to guarantee the bacteria clearance. In this context, to select the most competent RFB liposomal formulation, in terms of biofilm penetration, a transwell model was assembled (Figure 4d). Firstly, the ability of the strain under study to form a mature biofilm on the apex of the transwell was evaluated demonstrating similar behavior to the susceptibility experiments previously performed. The integrity of MSSA biofilm structure after exposure to RFB in free form, and in unloaded and loaded rhodamine-labelled liposomes (LIP1, LIP2, LIP3, RFB LIP1, RFB LIP2 and RFB LIP3) was evaluated by a CV assay (Figure 4e). These results revealed that the biofilm integrity was preserved after incubation with LIP1 and LIP2 (negatively charged), while a slight decrease for LIP3 (positively charged) was observed. These data are in accordance with literature, as the presence of SA in the lipid composition promotes the establishment of electrostatic interactions between positively charged liposomes and negatively charged bacterial cell envelope. This can lead to *S. aureus* cell membrane disruption and consequently the reduction of *S. aureus* biofilm [22,23,71,72,73,74]. For the three lipid compositions incorporating the antibiotic (RFB LIP1, RFB LIP2 and RFB LIP3) and free RFB a decrease on the biofilm biomass was verified. Interestingly, RFB LIP3 presented the highest biofilm biomass value (60%), with a statistically significant difference with a *p* < 0.0001 in comparison to free RFB, as shown in Figure 4e. The biofilm biomass reduction in relation to respective unloaded formulations is presented in Table 4. RFB LIP1 and RFB LIP2 displayed a biofilm reduction of 72 and 64%, respectively, and a lower reduction was observed for RFB LIP3 (32%). In fact, Scriboni et al. evaluated the susceptibility of MSSA and MRSA strains to VCM-loaded in positively charged and in fusogenic liposomes, proving the higher efficacy of VCM for the latest lipid composition towards mature biofilm [12]. Moreover, all RFB labelled liposomes presented similar antibiotic concentrations. The amount of lipid that remained or crossed the biofilm structure (designated by biofilm entrapped liposome, BEL) was quantified spectroflourimetrically, by measuring the rhodamine signal, at samples collected from the apex (Table 4). Similar to the results previously obtained (Section 3.4), RFB LIP3 exhibited the highest biofilm interaction levels, with a BEL value of 40%. The negatively charged liposomes achieved 17% and 23% of BEL, respectively. These data are supported by literature, describing that liposomes with surface charge opposite to that of bacteria might allow extended contact times (reviewed in [11,23]). In this case, the high level of RFB LIP3 entrapment into biofilm could be associated to its ionic interaction with negatively charged surface of the bacteria under study [12,23,44,75].

Due to the weak signal of the sample collected at the base of the transwell, it was not possible to determine the fluorescence intensity of rhodamine-labelled liposomes which crossed the biofilm. An alternative approach was developed to evaluate if RFB loaded in liposomes were able to effectively cross the biofilm and the membrane. To prove this hypothesis, the antibacterial effect of the samples recovered from the base was evaluated against planktonic MSSA strain. The samples referring to the RFB-loaded formulations (RFB LIP1, RFB LIP2 and RFB LIP3) reduced the bacterial growth by more than 98% in relation to respective unloaded liposomal formulation. This experiment proved that RFB released from liposomes and free RFB were able to cross the biofilm and, consequently, the transwell membrane, retaining its antibacterial activity. The samples recovered from unloaded liposomes (LIP1, LIP2 and LIP3) did not interfere with bacterial growth.

Overall, with these series of in vitro assays, we can conclude that, although RFB LIP3 presented higher interaction levels with MSSA biofilm, they did not present a higher anti-biofilm effect, as demonstrated by biofilm integrity results (Figure 4e).

To corroborate the liposome-biofilm interaction results obtained in the previous assays (Section 3.4 and Section 3.5), a CSLM analysis was performed. CLSM images were acquired at the biofilm inner layer and at an orthogonal plane of 24 h old biofilms. MSSA biofilm was incubated with rhodamine-labeled liposomes, in the absence of incorporating RFB, at a lipid concentration of 1.5 µmol/mL during 4 h, and then sequentially stained with SYTO 9 (Figure 5a), as the nucleic acid-binding dye, that stains all *S. aureus* green (with intact and damaged membranes) in a population [38]. A quantitative evaluation of the fluorescence intensity signal in relation to control (untreated 24 h old biofilms) was also assessed (Figure 5b,c).

Confocal microscopy studies suggest that the accumulation of liposomes within biofilm was lipid composition-dependent, in accordance with the previous results (Section 3.4 and Section 3.5). As shown in Figure 5a (overlay—xzy), both LIP3 and RFB LIP3, the positively charged formulations were able to cross and accumulate into MSSA biofilm structure in a higher extent, in comparison with the remaining formulations tested. LIP2 presented a slight increase in comparison with LIP1, according to the results observed in biofilm transwell model. This phenomenon was observed in plane xyz (Figure 5b), while in plane xzy LIP1 and LIP2 maintained their similarity (Figure 5c). This microscopical analysis also leads to the conclusion that no differences were observed between unloaded and loaded liposomes, for the same lipid composition.

### 3.6. Preliminary In Vitro Safety Assessment of RFB Liposomes

*S. aureus* biofilm infection is strongly associated with orthopedic surgeries due to its ability to adhere and form biofilms on bone and on implant surfaces [7]. Considering this, it is crucial to evaluate the biological response for a specific cell type of relevance for the proposed applications. In case of local administration of formulations, to treat this type of infection, osteoblast cell lines are usually chosen [7,76]. For a more generic assessment of biological response, fibroblasts are commonly used as a healthy cell line [7]. Taking this into account, the safety of RFB-loaded and -unloaded liposomes was conducted with human MG-63 osteoblast and mouse L929 fibroblast cell lines (Figure 6 and Appendix A). The cell viability was evaluated 24 h after incubation with selected RFB liposomes, at four lipid concentrations: 0.5, 1.0, 2.0 and 5.0 µmol/mL. The obtained data depicted in Figure 6 revealed that osteoblast cell viability was not compromised after incubation with negatively charged formulations RFB LIP1 and RFB LIP2 (Figure 6a). However, a slight cell viability reduction was observed for RFB LIP1, particularly for higher tested lipid concentrations towards the fibroblast cell line (Figure 6b).

On the other hand, the positively charged formulation containing SA in the lipid composition (RFB LIP3) promoted a huge decrease in cell viability for osteoblast and fibroblast cell lines (Figure 6a,b). In all tested lipid concentrations, a cell viability reduction of approximately 70% was observed for both cellular populations incubated with RFB LIP3. Although positively charged liposomes were able to establish high levels of interaction/accumulation with *S. aureus* biofilms, proved by previously assays (Section 3.4 and Section 3.5), this preliminary in vitro safety assay may limit their in vivo application. Indeed, the in vitro toxic effects of positively charged liposomes containing SA in the lipid composition is described in literature [23,77,78,79,80]. In vitro studies performed with macrophages showed that this type of formulations induces apoptosis through mitochondrial pathways, producing reactive oxygen species, activating protein kinase C, caspase-3 and -8 and releasing cytochrome c, playing important roles in the regulation of several cellular processes [78,79,80]. This result can also explain the slight reduction of biofilm biomass obtained when *S. aureus* biofilm was incubated with LIP3 formulation, in the biofilm transwell model experiment (Section 3.5). Furthermore, all unloaded liposomes, LIP1, LIP2 and LIP3, showed similar behavior as the respective RFB-loaded formulations: RFB LIP1, RFB LIP2 and RFB LIP3 (Appendix A), demonstrating that the obtained results depended only on the lipid composition used.

## 4. Conclusions

In this study, we developed three antibiotic-loaded liposomes (RFB, LEV and VCM) with improved interaction and accumulation into biofilm organized MSSA, correlating with their antibacterial activity. Among the antibiotics under study, RFB stood out regarding its antibacterial effect, both against the planktonic and biofilm forms of *S. aureus* strain tested. To the best of our knowledge, this is the first study in which RFB-loaded liposomes were developed as a potential therapy platform for staphylococcal biofilm infections. Taking this into account, the antibacterial and anti-biofilm activity of RFB-loaded liposomes was validated by in vitro susceptibility tests using a MSSA strain. Susceptibility studies demonstrated that RFB incorporated in all liposomal formulations tested (negatively charged liposomes, with or without fusogenic properties, and positively charged liposomes) preserved its antibacterial activity, however, with slightly MIC value variation, depending on the lipid composition. Although our results showed the high interaction level of positively charged liposomes with MSSA biofilm, this was not translated into an improved anti-biofilm efficacy. Probably, the highest interaction of positively charged liposome with biofilm structure restrains the release of entrapped antibiotic. Furthermore, in preliminary in vitro tests, this lipid composition promoted a huge reduction on cell viability. On the contrary, negatively charged RFB formulations after incubation with osteoblast and fibroblast cell lines, did not interfere with cell viability. The results obtained in this study revealed the pivotal importance of the design and optimization of antibiotic-loaded liposomes with appropriate features able to penetrate and accumulate into biofilms and successfully release the antibiotic in situ. In conclusion, this nanotechnological strategy associated to RFB, particularly with negatively charged liposomes, presents a high therapeutic potential thus constituting a promising approach to promote safer and more effective treatments for biofilm *S. aureus* infections.

## Figures and Tables

**Figure 1 pharmaceutics-13-00321-f001:**
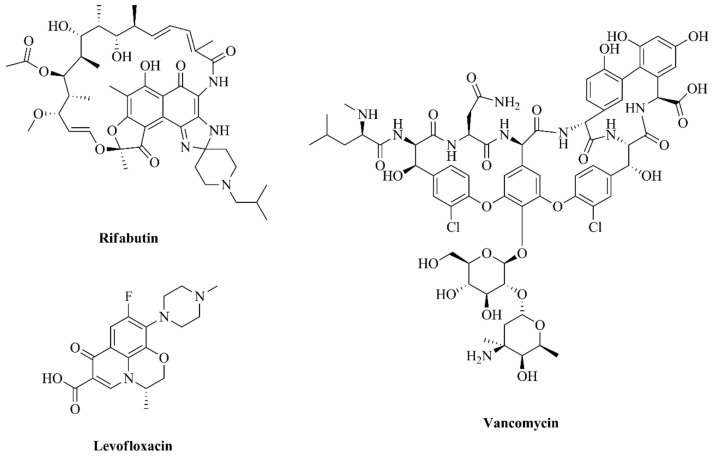
Chemical structures of antibiotics selected in the current study: rifabutin (RFB), levofloxacin (LEV) and vancomycin (VCM). Chemical structures were obtained through a chemistry drawing software, ChemDraw^®^ ultra 12.0.

**Figure 2 pharmaceutics-13-00321-f002:**
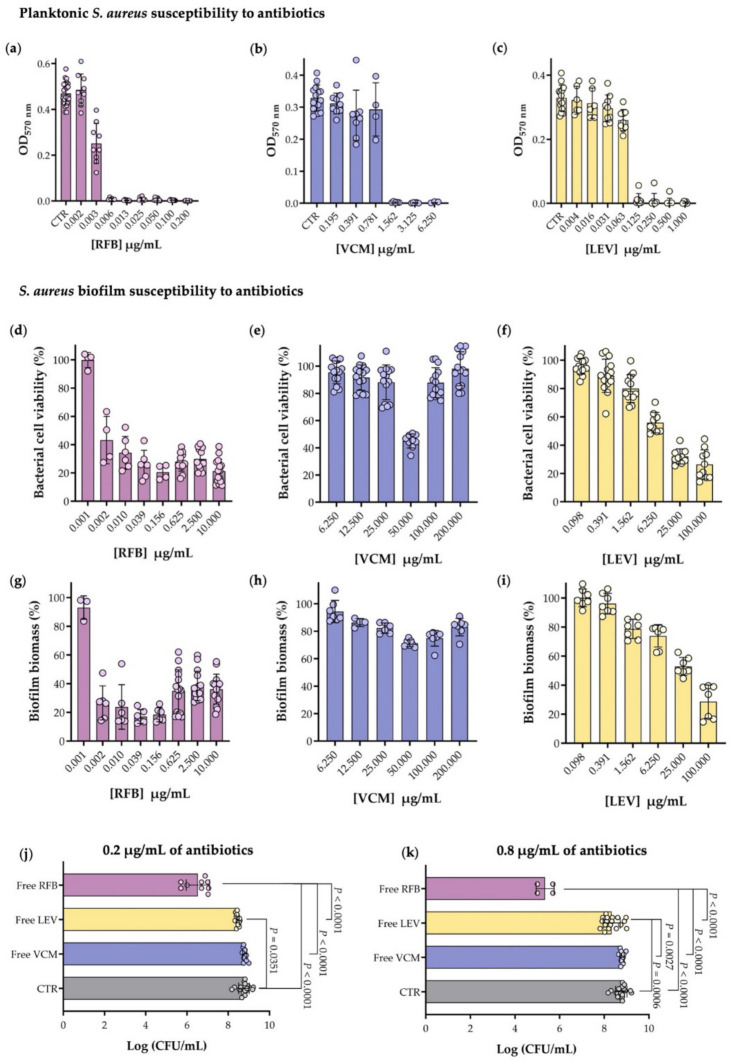
(**a**–**c**) In vitro planktonic MSSA susceptibility to free antibiotics assessment through broth microdilution method followed by turbidity measurement (OD_570 nm_), after 24 h of incubation. The negative control corresponds to planktonic MSSA in MHB (CTR). Antibiotic concentrations of (**a**) RFB (0.002–0.200 µg/mL), (**b**) VCM (0.195–6.250 µg/mL) and (**c**) LEV (0.004–1.000 µg/mL); (**d**–**i**) MSSA biofilm susceptibility to free antibiotics was evaluated through the broth microdilution method, performed in mature biofilms (24 h old), followed by determination of (**d**–**f**) viable bacterial cells (MTT assay) and by (**g**–**i**) biofilm biomass quantification (CV method), after 24 h of incubation. Bacterial cell viability (%) in biofilm structure after incubation (**d**) with RFB (0.001–10.000 µg/mL), (**e**) with VCM (6.250–200.000 µg/mL) and (**f**) with LEV (0.098–100.000 µg/mL). Biofilm biomass (%) present after incubation (**g**) with RFB (0.001–10.000 µg/mL), (**h**) with VCM (6.250–200.000 µg/mL) and (**i**) with LEV (0.098–100.000 µg/mL). (**j**,**k**) Viable bacteria recovered from mature biofilm structure after 24 h of incubation with the selected antibiotics, RFB, VCM and LEV, were determined by CFU counts. Statistical comparisons were assayed by one-way ANOVA (Tukey’s multiple comparisons test) analysis of variance compared with control group (CTR, biofilm in TSB 0.25%) and between antibiotic groups. (**j**) Biofilm incubated with 0.2 µg/mL of each antibiotic. (**k**) Biofilm incubated with 0.8 µg/mL of each antibiotic. Results are expressed as mean ± SD of at least three independent experiments.

**Figure 3 pharmaceutics-13-00321-f003:**
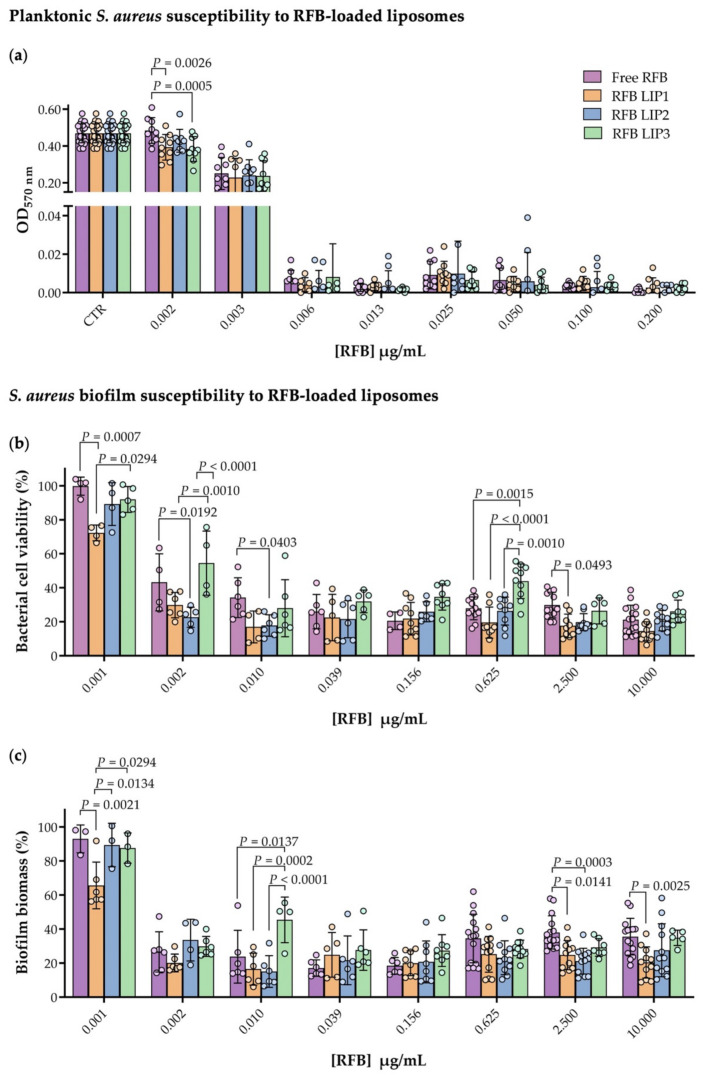
(**a**) In vitro planktonic MSSA susceptibility to free and liposomal RFB assessment through broth microdilution method, followed by turbidity measurement (OD_570 nm_), RFB concentrations tested ranged from 0.002 to 0.200 µg/mL. The negative control corresponds to planktonic MSSA in MHB (CTR). (**b**) Determination of viable bacterial cells (MTT assay) and (**c**) biofilm biomass quantification (CV method) after mature MSSA biofilm incubation with a RFB concentration ranging from 0.001 to 10.000 µg/mL. Statistical comparisons were determined by two-way ANOVA (Tukey’s multiple comparisons test) analysis of variance compared between formulation groups. Results are expressed as mean ± SD of at least three independent experiments.

**Figure 4 pharmaceutics-13-00321-f004:**
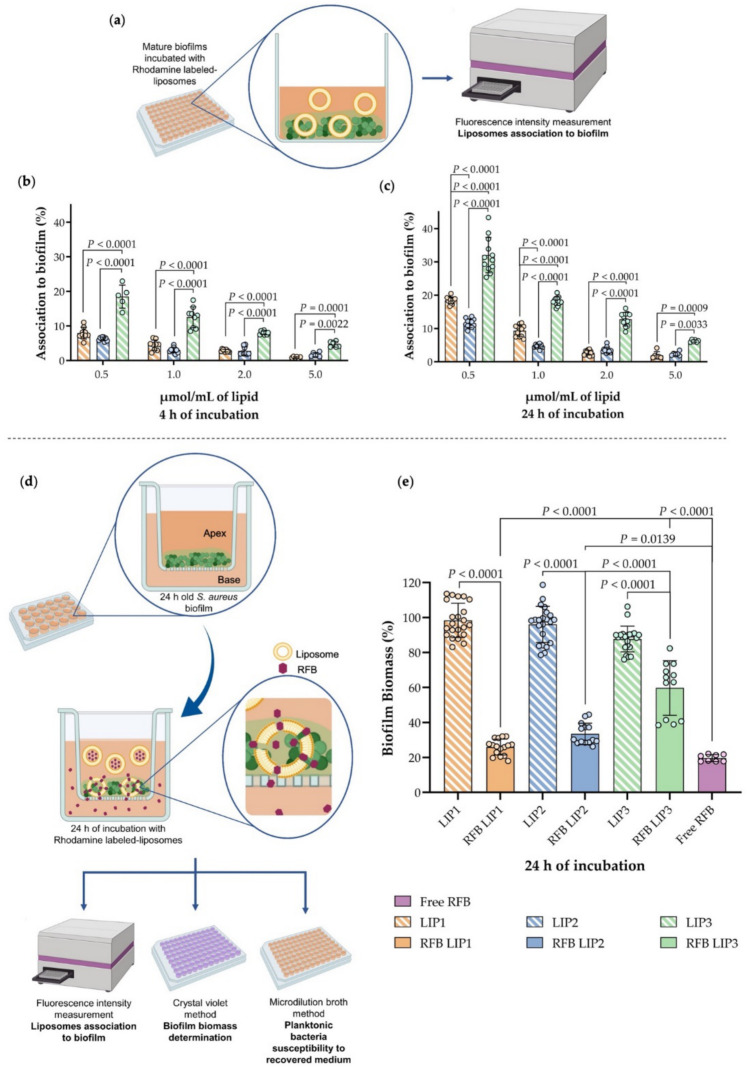
(**a**) Schematic representation of the experimental protocol for the liposome-biofilm interaction evaluation, performed in 96-well cell culture plate with rhodamine labeled-liposomes (LIP1, LIP2 and LIP3), at 0.5, 1, 2 and 5.0 µmol/mL of lipid, against mature MSSA biofilm. The presence of liposomes was evaluated by fluorescence intensity measurement (Spectrofluorimetry, λ excitation = 540 nm; λ emission = 620 nm); (**b**) Incubation for 4 h and (**c**) incubation for 24 h. Results are expressed as mean ± SD of at least three independent experiments. Statistical comparisons were determined by two-way ANOVA (Tukey’s multiple comparisons test) analysis of variance compared between formulation groups. (**d**) Schematic representation of biofilm transwell model. Biofilm grew in a porous membrane cell culture insert (apex), placed on 24-well cell culture plate (base). RFB-loaded and unloaded in rhodamine labelled-liposomes (RFB LIP1, RFB LIP2, RFB LIP3, LIP1, LIP2, LIP3,) were incubated with mature MSSA biofilm in the apex, for 24 h. Liposomes internalization was evaluated by spectrofluorimetry (λ excitation = 540 nm; λ emission = 620 nm), biofilm biomass was determined by CV assay and planktonic susceptibility in the recovered medium by broth microdilution method. (**e**) Biofilm biomass (%) quantification of biofilm transwell model. Results are expressed as mean ± SD of two independent experiments. Statistical comparisons were determined by one-way ANOVA (Bonferroni’s multiple comparisons test) analysis of variance compared between formulation groups.

**Figure 5 pharmaceutics-13-00321-f005:**
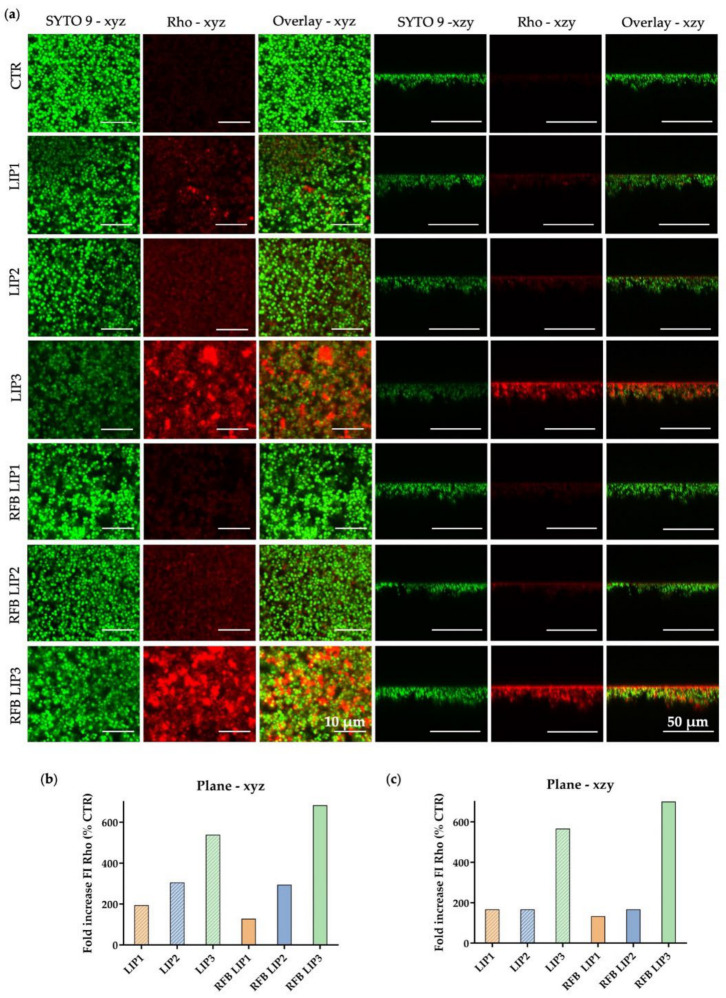
(**a**) Representative CLSM images of 24 h old biofilms incubated with unloaded or RFB-loaded liposomes labelled with Rho (LIP1, LIP2, LIP3, RFB LIP1, RFB LIP2 and RFB LIP3) at a lipid concentration of 1.5 µmol/mL, during 4 h; Untreated 24 h old biofilm was used as a control (CTR). The biofilms were stained with the nucleic acid-binding dye (SYTO 9) at 3 µM, (green). The panels of the left correspond to xyz plane images taken at the inner layer of MSSA stained biofilms and the right panels consist of xzy orthogonal plane images. The overlay of the green and red channels from xyz and xzy plane images is presented in the central and right panels, respectively. (**b**,**c**) Evaluation of the fluorescent intensity (FI) increase in relation to CTR, (**b**) in plane xyz and (**c**) in plane xzy.

**Figure 6 pharmaceutics-13-00321-f006:**
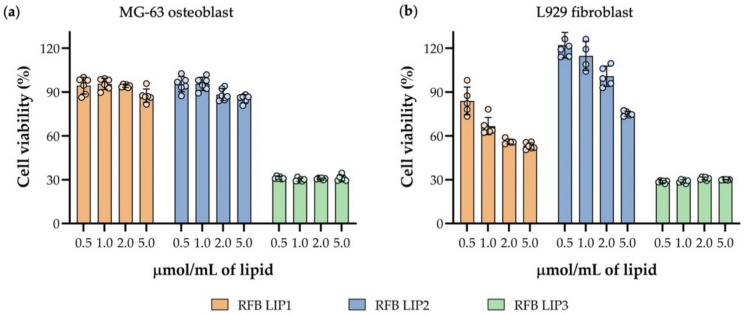
Cell viability of (**a**) human MG-63 osteoblast and (**b**) mouse L929 fibroblast cell lines 24 h after incubation with RFB-loaded liposomes (RFB LIP1, RFB LIP2 and RFB LIP3), at a lipid concentration of 0.5, 1.0, 2.0 and 5.0 µmol/mL. Cell viability was determined by MTT reduction assay. Results are expressed as mean ± SD of at least two independent experiments.

**Table 1 pharmaceutics-13-00321-t001:** Planktonic and biofilm MSSA susceptibility to the three selected antibiotics.

Antibiotics	MIC ^1^ (µg/mL)	MBIC_50_ ^2^ (µg/mL)
RFB	0.006 ± 0.000	0.005 ± 0.002
VCM	1.562 ± 0.033	>200.000
LEV	0.125 ± 0.068	9.468 ± 0.672

^1^ The lowest antibiotic concentration able to prevent visible growth after 24 h of incubation. ^2^ The lowest antibiotic concentration able to inhibit more than 50% of biofilm growth after 24 h of incubation, determined by MTT assay.

**Table 2 pharmaceutics-13-00321-t002:** Physicochemical properties of antibiotics-loaded liposomes: influence of the antibiotic and lipid composition on incorporation parameters.

Antibiotics	Lipid Composition (Molar Ratio)	Loading Capacity (µg/µmol)	I.E. (%)	Ø (µm) (P.I.)	Zeta Potential (mV)
RFB	DMPC:DOPE:CHEMS (4:4:2)	57 ± 9	87 ± 5	0.11 (<0.10)	−22 ± 3
	DMPC:DMPG (8:2)	36 ± 5	51 ± 7	0.15 (<0.10)	−21 ± 3
	DMPC:SA (9:1)	24 ± 4	32 ± 3	0.12 (<0.10)	+13 ± 2
VCM	DPPC:DOPE:CHEMS (4:4:2)	45 ± 3	19 ± 4	0.17 (<0.10)	−30 ± 1
	DPPC:DPPG (8:2)	23 ± 2	32 ± 8	0.15 (<0.15)	−23 ± 1
LEV	DMPC:DOPE:CHEMS (4:4:2)	<2	<3	0.12 (<0.10)	−21 ± 2
	DMPC:DMPG (8:2)	<2	<3	0.11 (<0.10)	−24 ± 1
	DMPC:SA (9:1)	<2	<3	0.13 (<0.10)	+18 ± 1

Initial lipid concentration, [Lip]i—30 µmol/mL; Initial antibiotic concentration, (RFB/Lip)i—100 nmol RFB/µmol of lipid, [VCM]i—2 mg/mL, [LEV]i—1 mg/mL; AB—Antibiotic; Loading capacity—(AB/Lip)f (µg/µmol); I.E. (%)—Incorporation Efficiency, [(AB/Lip)f]/[(AB/Lip)i] × 100; Ø—mean size; P.I.—polydispersity index; DPPC—dipalmitoyl phosphatidyl choline; DPPG—dipalmitoyl phosphatidyl glycerol; CHEMS—cholesteryl hemisuccinate; DMPC—dimyristoyl phosphatidyl choline; DMPG—dimyristoyl phosphatidyl glycerol; SA—stearylamine; DOPE—dioleoyl phosphatidyl ethanolamine; Results are expressed as mean ± SD of at least three independent experiments.

**Table 3 pharmaceutics-13-00321-t003:** Planktonic and biofilm MSSA susceptibility to RFB in free form or loaded in liposomes.

Formulation	MIC ^1^ (µg/mL)	MBIC_50_ ^2^ (µg/mL)
Free RFB	0.006 ± 0.000	0.005 ± 0.0032
LIP1	0.006 ± 0.004	0.002 ± 0.0001
LIP2	0.006 ± 0.000	0.002 ± 0.0002
LIP3	0.006 ± 0.004	0.006 ± 0.0030

^1^ The lowest RFB concentration able to prevent visible growth after 24 h of incubation. ^2^ The lowest RFB concentration able to inhibit more than 50% of biofilm growth, after 24 h of incubation, determined by MTT assay.

**Table 4 pharmaceutics-13-00321-t004:** RFB-loaded liposomes interaction with mature MSSA biofilm: biofilm transwell model.

Lipid Composition	BEL (%)	Biofilm Biomass Reduction (%) ^1^
RFB LIP1	17 ± 9	72 ± 5
RFB LIP2	23 ± 6	64 ± 9
RFB LIP3	40 ± 8	32 ± 14

BEL—Biofilm entrapped liposome. ^1^ Biofilm biomass reduction in relation to the respective unloaded liposomal formulation.

## Data Availability

The data presented in this study are available on request from the corresponding author.

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
