# Peer review of "Liposomes as a Nanoplatform to Improve the Delivery of Antibiotics into Staphylococcus aureus Biofilms"

_pharmaceutics, 2021, doi:10.3390/pharmaceutics13030321_

Round 1

Reviewer 1 Report

In this work the authors firstly assessed the antimicrobial effect of three antibiotics, vancomycin, levofloxacin and rifabutin and subsequently they developed different rifabutin-containing liposomal formulations to evaluate the influence of lipid composition of liposomes on the ability to interact with Staphylococcus aureus biofilm. Additionally, the safety of liposomes was investigated towards osteoblast and fibroblast cell lines.

The work is interesting and meaningful and it is generally well-written. But, some questions should be clarified or corrected in the presented manuscript before publication. I suggest minor revisions before acceptance. Some detailed issues are as follows:

Abstract: please explain the abbreviation RFB.

Line 493: the authors reported the following sentence “The higher fluidity of the lipid composition, as a result of the presence of DOPE, allows a higher penetration through the lipid bilayer of RFB, following their incubation with empty liposomes and their precipitation in the internal aqueous compartment”. Please clarify this sentence. Considering the physico-chemical properties of RFB, the authors should better explain and discuss where the drug is located (in the internal aqueous compartment or also in the lipid bilayer?)

Line 497: the authors reported the following sentence “ although negatively charged displayed lower loadings.” The authors should explain the main factors related to this result.

Lines 498- 505: the authors should deeply discuss results related to zeta potential (especially the negative zeta potential of  liposomes without SA). Moreover, what are the implications of zeta potential results on liposome stability?

The authors should provide data regarding  the morphology of liposomes.

Lines 542-544: considering that RFB LIP3 showed the highest antibacterial acitivity among all the liposomes with different composition (containing the same drug concentration), probably a synergistc effect between RFB and lipidic composition could explain the results.

Line 568: “LIP3 showed a slightly higher MBIC50, 0.006 μg/mL.” Please include p value in the text.

Author Response

In this work the authors firstly assessed the antimicrobial effect of three antibiotics, vancomycin, levofloxacin and rifabutin and subsequently they developed different rifabutin-containing liposomal formulations to evaluate the influence of lipid composition of liposomes on the ability to interact with Staphylococcus aureus biofilm. Additionally, the safety of liposomes was investigated towards osteoblast and fibroblast cell lines.

The work is interesting and meaningful and it is generally well-written. But, some questions should be clarified or corrected in the presented manuscript before publication. I suggest minor revisions before acceptance. Some detailed issues are as follows:

  • Abstract: please explain the abbreviation RFB.

Reply – The meaning of the abbreviation, RFB, (rifabutin) was included in the abstract.

  • Line 493: the authors reported the following sentence “The higher fluidity of the lipid composition, as a result of the presence of DOPE, allows a higher penetration through the lipid bilayer of RFB, following their incubation with empty liposomes and their precipitation in the internal aqueous compartment”. Please clarify this sentence. Considering the physico-chemical properties of RFB, the authors should better explain and discuss where the drug is located (in the internal aqueous compartment or also in the lipid bilayer?)

Reply – RFB is located in the internal aqueous compartment of liposomes. As described in methods section, RFB was actively incorporated in pre-formed empty liposomes in response to a salt gradient; in the present case an ammonium sulphate gradient between intraliposomal and extraliposomal media. This active method is based on the high permeability of neutral molecules across lipid membranes in preformed liposomes and on the other hand on the very low permeation of charged molecules. In this sense, empty liposomes were incubated with a RFB solution, prepared in HEPES buffer, pH 6,9, the pKa value for RFB. This means that the higher fluidity of the lipid bilayer, the easier will be for RFB to pass through the lipid bilayer. On the other hand, the stability of the ammonium ion gradient is related to the low permeability of its counterion, the sulphate, which also stabilizes RFB accumulation that gets charged after reaching aqueous compartment of liposomes and due to that is not able to pass thorough the lipid bilayer getting sequestered in the internal aqueous phase.  

The following sentence and one reference were included in the revised manuscript: The higher fluidity of the lipid composition, as a result of the presence of DOPE, allows a higher penetration through the lipid bilayer of RFB, that is incubated in uncharged state with empty liposomes, being able to cross through the lipid bilayers and when reaches the aqueous compartment gets charged keeping sequestered in the internal aqueous phase. The same methodology has been widely used for efficient and stable active loadings of amphipathic weak bases in liposomes (Bolotin et al. 1994).

  • Line 497: the authors reported the following sentence “ although negatively charged displayed lower loadings.” The authors should explain the main factors related to this result.

Reply - Thank you for the observation. The sentence “For VCM liposomes, the more rigid lipid composition, DPPC:DPPG, (with a Tc of 41 °C) enabled to achieve higher loadings for that was incorporated by a passive method, although negatively charged displayed lower loadings.”

Was changed to:

“For the incorporation of the hydrophilic glycopeptide antibiotic, VCM, a passive method was used. Two negatively charged lipid formulations were tested. Again as observed for RFB, the more fluid lipid composition containing DOPE and CHEMS displayed higher loadings in comparison to DPPC:DDPG, phospholipids with a Tc of 41 °C. “

  • Lines 498- 505: the authors should deeply discuss results related to zeta potential (especially the negative zeta potential of  liposomes without SA). Moreover, what are the implications of zeta potential results on liposome stability?

Reply- Two additional sentences were included.

The negatively charged observed for liposomes is in accordance with the lipids used in the lipid composition, namely DMPG, DPPG and CHEMS. From a general point of view, charged liposomes are more stable as the presence of a charge on the surface induces electrostatic repulsion among liposomes allowing also to promote the interaction of liposomes with cells (Bozzuto and Molinari 2015).”

  • The authors should provide data regarding the morphology of liposomes.

Reply – We have not included in the manuscript a figure demonstrating the morphology of liposomes. Nevertheless, the dynamic light scattering methodology that was extensively performed for characterizing the mean size and the polydispersity index of developed liposomes clearly evidences their high homogeneity.

  • Lines 542-544: considering that RFB LIP3 showed the highest antibacterial activity among all the liposomes with different composition (containing the same drug concentration), probably a synergistc effect between RFB and lipidic composition could explain the results.

Reply – In our opinion, we think that a synergistic effect between RFB loaded and unloaded liposomes was not observed. In fact, in Supplementary material, Figure S1a is shown that RFBLIP3 and LIP3 presented similar absorbance values at 570 nm for the lowest RFB concentration tested (RFBLIP3) (0.002 µg/ml) and the correspondent lipid concentration for unloaded liposomes (LIP3). We removed the sentence “Moreover, RFB LIP3 displayed the highest antibacterial effect” as the MIC for RFB in free form or loaded in liposomes was the same (Table 3 – 0.006 µg/ml).

  • Line 568: “LIP3 showed a slightly higher MBIC50, 0.006 μg/mL.” Please include p value in the text.

Reply – the p value was included in the text: “p>0.05

Reviewer 2 Report

The authors present an interesting study describing the preparation and optimization of a liposome-based platform for antibiotic delivery. The manuscript is nicely written being the results feasible and rigorously presented. It is worth publishing it in Pharmaceutics. There are few doubts/comments arising from the reading.

  1. Liposomes were prepared by the dehydration-rehydration method, firstly described by Kirby and Gergoriadis in 1984, therefore these authors should be cited.
  2. If the liposomes are extruded after rehydration, is it really needed the first step?
  3. RFB is incorporated into liposomes by a gradient procedure. Being a hydrophobic compound, why do not incorporate it to the bilayer during the film step?
  4. Positively charged lipids are toxic as checked in the cellular studies. Do the authors believe it would be suitable a preparation of these characteristics for human use?
  5. Cytotoxicity of control liposomes is mentioned in the text but not shown in the figures.
  6. Self-citation reaches a 22 %.

Author Response

The authors present an interesting study describing the preparation and optimization of a liposome-based platform for antibiotic delivery. The manuscript is nicely written being the results feasible and rigorously presented. It is worth publishing it in Pharmaceutics. There are few doubts/comments arising from the reading.

  1. Liposomes were prepared by the dehydration-rehydration method, firstly described by Kirby and Gergoriadis in 1984, therefore these authors should be cited.

Reply - The paper from Prof. Gregoriadis was included in methods section (Preparation of VCM- and LEV-loaded liposomes)

2. If the liposomes are extruded after rehydration, is it really needed the first step?

Reply – According to the question of the reviewer if we understood you are asking why the rehydration is performed in two steps.

It is assumed that during dehydration, vesicles become flatter and fuse forming multilamellar planes in which the solute is sandwiched allowing the achievement of higher entrapments than those obtained by the thin film hydration method. In addition, the first rehydration step, which is performed using two tenth of the original solution volume results in a five-fold increase in overall concentration of the solute, being this reflected in the concentration of the material that is actually entrapped. Overall, the drying process (lyophilization) brings the bilayers and material to be encapsulated into very close contact and upon reswelling, the chances for loading molecules are higher (Lasch et al. 2003).

3. RFB is incorporated into liposomes by a gradient procedure. Being a hydrophobic compound, why do not incorporate it to the bilayer during the film step?

Reply – In the first published work using rifabutin liposomes (Gaspar et al. 2000), a passive method for incorporating rifabutin in liposomes was used. However, following studies, using an active method was performed (Gaspar et al. 2008) and higher loadings and stability were observed for rifabutin nanoformulations prepared in pre-formed empty liposomes in response to a salt gradient.

4. Positively charged lipids are toxic as checked in the cellular studies. Do the authors believe it would be suitable a preparation of these characteristics for human use?

Reply – Although some toxic effects are attributed to the use of positively charged formulations, particularly in those including stearylamine, these effects are dose dependent. On the other hand, this effect may be therapeutical advantageous: it has been reported that liposomes including SA in the lipid composition exhibited antileishmanial activity, possibly through an interaction of the positively charged lipids with the negatively charged parasite membrane (Dey et al. 2000). Regarding S. aureus infections, the same effect has been described in literature and confirmed in the present work. Preliminary in vitro safety assessment achieved in the present work also demonstrated that. A balance between therapeutic effect and toxic side effects must be assessed. In vivo studies regarding the safety and the therapeutic properties of negatively and positively charged liposomes should be conducted.

5. Cytotoxicity of control liposomes is mentioned in the text but not shown in the figures.

Reply – If we understood the question of the reviewer, in Figure 6 of the manuscript, is shown the cell viability of osteoblasts and fibroblasts after incubation with RFB liposomes and in supplementary material, Figure S3, is depicted the in vitro safety results using the same cell lines after incubation with unloaded liposomes and the same lipid concentrations as those used in Figure 6.

6. Self-citation reaches a 22 %.

Reply – the reason of self-citation means that the area of research of the co-authors has not begun recently, but represents a continuous investigation of several decades.

References:

Gaspar, M.M., A. Cruz, A.F. Penha, J. Reymão, A.C. Sousa, C.V. Eleutério, S.A. Domingues, et al. 2008. “Rifabutin Encapsulated in Liposomes Exhibits Increased Therapeutic Activity in a Model of Disseminated Tuberculosis.” International Journal of Antimicrobial Agents 31 (1): 37–45. https://doi.org/10.1016/j.ijantimicag.2007.08.008.

Gaspar, M M, S Neves, F Portaels, J Pedrosa, M T Silva, and M E Cruz. 2000. “Therapeutic Efficacy of Liposomal Rifabutin in a Mycobacterium Avium Model of Infection.” Antimicrobial Agents and Chemotherapy 44 (9): 2424–30. https://doi.org/10.1128/aac.44.9.2424-2430.2000.

Lasch, J, V Weissig, and M Brandl. 2003. “Preparation of Liposomes.” In Liposomes a Pratical Approach, edited by V.P. Torchillin and V Weissig, 3–29. New York: Oxford University Press.

Dey, T, Anam, K, Afrin, F, Ali, N., 2000. “Antileishmanial Activities of Stearylamine-Bearing Liposomes”. Antimicrob Agents Chemother 44 (6): 1739-42. https://doi: 10.1128/aac.44.6.1739-1742.2000.

Reviewer 3 Report

The evaluated manuscript has the character of the original work. 

The chosen topic is highly actual and brings another possibility to solve the problem of bacterial resistance to antibiotics.

The presented manuscript is processed very comprehensively with high quality. I find the methodologies used to be very adequate and I would like to emphasize their appropriate selection.

The results are very convincing and the discussion is optimal.

The high potential of this article to be cited can be assumed.

I have no comments and I would like to congratulate the authors on this work.

Author Response

The evaluated manuscript has the character of the original work. 

The chosen topic is highly actual and brings another possibility to solve the problem of bacterial resistance to antibiotics.

The presented manuscript is processed very comprehensively with high quality. I find the methodologies used to be very adequate and I would like to emphasize their appropriate selection.

The results are very convincing and the discussion is optimal.

The high potential of this article to be cited can be assumed.

I have no comments and I would like to congratulate the authors on this work.

Reply – the authors would like to thank the comments of the reviewer